# Germline mutations in mitochondrial complex I reveal genetic and targetable vulnerability in IDH1-mutant acute myeloid leukaemia

Mahmoud A. Bassal [1,2,14], Saumya E. Samaraweera [3,14], Kelly Lim [4], Brooks A. Benard [5], Sheree Bailey [6], Satinder Kaur[4], Paul Leo[7], John Toubia [3], Chloe Thompson-Peach [4], Tran Nguyen[3], Kyaw Ze Ya Maung[3], Debora A. Casolari [3], Diana G. Iarossi[3], Ilaria S. Pagani[8], Jason Powell[3], Stuart Pitson [3], Siria Natera[9], Ute Roessner [9], Ian D. Lewis[10], Anna L. Brown [3,6,11], Daniel G. Tenen[2,1], Nirmal Robinson [3], David M. Ross[3,4,8,12], Ravindra Majeti [5], Thomas J. Gonda[6,13], Daniel Thomas [4,5,8] & Richard J. D'Andrea [3✉]

The interaction of germline variation and somatic cancer driver mutations is under-investigated. Here we describe the genomic mitochondrial landscape in adult acute myeloid leukaemia (AML) and show that rare variants affecting the nuclear- and mitochondrially-encoded complex I genes show near-mutual exclusivity with somatic driver mutations affecting isocitrate dehydrogenase 1 (*IDH1*), but not *IDH2* suggesting a unique epistatic relationship. Whereas AML cells with rare complex I variants or mutations in *IDH1* or *IDH2* all display attenuated mitochondrial respiration, heightened sensitivity to complex I inhibitors including the clinical-grade inhibitor, IACS-010759, is observed only for *IDH1*-mutant AML. Furthermore, *IDH1* mutant blasts that are resistant to the IDH1-mutant inhibitor, ivosidenib, retain sensitivity to complex I inhibition. We propose that the *IDH1* mutation limits the flexibility for citrate utilization in the presence of impaired complex I activity to a degree that is not apparent in *IDH2* mutant cells, exposing a mutation-specific metabolic vulnerability. This reduced metabolic plasticity explains the epistatic relationship between the germline complex I variants and oncogenic *IDH1* mutation underscoring the utility of genomic data in revealing metabolic vulnerabilities with implications for therapy.

[1] Harvard Stem Cell Institute, Harvard Medical School, Boston, USA. [2] Cancer Science Institute of Singapore, National University of Singapore, Singapore, Singapore. [3] Centre for Cancer Biology, University of South Australia and SA Pathology, Adelaide, Australia. [4] Discipline of Medicine, University of Adelaide, Adelaide, Australia. [5] Hematology Division, Department of Medicine, Stanford Cancer Institute, Institute for Stem Cell and Regenerative Medicine, Stanford University, Stanford, USA. [6] Clinical and Health Sciences, University of South Australia, Adelaide, Australia. [7] Diamantina Institute, Translational Research Institute, Brisbane, Australia. [8] Precision Medicine Theme, South Australian Health and Medical Research Institute, Adelaide, Australia. [9] Metabolomics Australia, The University of Melbourne, Melbourne, Australia. [10] Adelaide Oncology & Haematology, Adelaide, Australia. [11] Department of Genetics and Molecular Pathology, SA Pathology, Adelaide, SA, Australia. [12] Department of Clinical Haematology, Royal Adelaide Hospital, Adelaide, Australia. [13] School of Pharmacy, University of Queensland, Brisbane, Australia. [14] These authors contributed equally: Mahmoud A. Bassal, Saumya E. Samaraweera. ✉email: Richard.DAndrea@unisa.edu.au

While it has long been recognized that tumour cells utilize glycolysis even under aerobic conditions[1], it is now clear that mitochondria and oxidative phosphorylation (OXPHOS) are also essential contributors to tumour growth and viability[2]. In addition to ATP generation, OXPHOS is critical for the oxidation of ubiquinol, maintenance of the tricarboxylic acid (TCA) cycle, aspartate, and pyrimidine synthesis, all of which are rate-limiting for tumour growth[3–7]. Such studies highlight the crucial role, independent of ATP production, of mitochondrial respiration in tumour growth, and the potential of OXPHOS as a cancer therapeutic target[3,4,8,9]. Furthermore, they emphasize the importance of identifying tumour subtypes that display unique dependencies and are responsive to clinical grade OXPHOS inhibitors that are reported to have selective anti-tumour activity[10–12]. AML is a highly heterogenous malignancy and there is variable dependency on OXPHOS across different subtypes[10]. Given data showing association of OXPHOS with AML chemoresistance and relapse[13]; there is much recent interest in the application of therapeutic approaches that target OXPHOS[14]. A major challenge for therapeutic targeting of OXPHOS is that tumours display remarkable metabolic reprogramming capacity ("metabolic plasticity") allowing adaptive survival under a variety of stresses and conditions to maintain ATP production, NADPH levels providing critical protection from reactive oxygen species (ROS), and biosynthesis of macromolecules to support uncontrolled growth[15,16]. Most recently, transient metabolic adaption in the face of chemotherapy has been demonstrated to provide a mechanism of escape for AML cells, leading to relapse[9]. Such non-mutational, metabolic mechanisms have also been shown to contribute to persistence of rare cell populations that proliferate after anti-cancer therapy in solid cancers[17]. We speculated that the capacity for such metabolic tumour plasticity may be in part determined by rare genetic variants affecting the mitochondrial respiratory chain (MRC). While rare variants affecting the mitochondrially encoded MRC complex I (NADH:CoQ oxidoreductase) genes have been described in cancer[18–20] and recently in AML[21], a comprehensive analysis of rare variants affecting the 86 nuclear-encoded MRC genes has not been performed in cancer cohorts. Here we show that rare variants affecting both mitochondrially encoded and nuclear-encoded complex I genes display near-mutual exclusivity with somatically acquired mutations in *IDH1*, but not *IDH2*. AML cells with rare complex I variants, or mutations in *IDH1* or *IDH2*, display attenuated mitochondrial respiration, however heightened sensitivity to IACS-010759 is only observed for *IDH1*-mutant AML. *IDH1* mutant blasts that are resistant to the IDH1-mutant inhibitor, ivosidenib, also retain sensitivity to complex I inhibition.

## Results

**Rare, nuclear-encoded, MRC variants in adult AML.** We investigated rare MRC genetic variants in an adult AML cohort, and integrated this with somatic driver mutations and metabolic profiles. We analysed data from a whole-exome sequencing study of 145 diagnostic AML samples[22] and identified all rare variants in the 86 nuclear-encoded MRC subunits (minor allele frequency ≤0.005, see the "Methods" section). MRC subunits show extremely high conservation[23], and assemble in a tightly integrated quaternary structure, suggesting that subtle sequence alterations may destabilize the macrostructure conferring a metabolic phenotype that contributes to cancer progression[24]; thus all rare variants were considered without filtering for predicted pathogenicity. Herein we use the term "variants" to define the rare variants identified from this analysis while the term "mutations" is used to refer to somatic driver mutations common

in AML (e.g., *IDH1* and *IDH2* mutations). We identified 139 rare variants in 94 patients (average of 1.5 variants per patient), distributed across all five complexes (Fig. 1a, Supplementary Data 1). 121 nuclear-encoded variants were observed at a variant allele frequency (VAF) >30%, compatible with germline mutation. Indeed, for 21 patients for whom we also had non-hematopoietic material available, we confirmed that 22 of 24 MRC variants were germline and only two were somatic (Supplementary Data 1), consistent with prior genome-wide studies of somatic mutations in AML[25]. Strikingly, when we examined the mutational profile of our cohort, including the common recurrent somatic mutations, AML cases harbouring rare variants in mitochondrial complex I were conspicuous for the absence of the *IDH1* R132 mutation ($P = 0.05$, Fisher's exact test, Supplementary Data 2 and Supplementary Figure 1a). Oncogenic, hotspot *IDH1* and *IDH2* mutations occur in 20% of AML cases and mechanistically it is well-established that these generate high levels of 2-hydroxyglutarate (2HG) that aberrantly inhibit the activity of multiple epigenetic regulators leading to differentiation block and oncogenesis[26,27]. There was no difference in the frequency of other recurrent mutations for samples with complex I variants (Supplementary Data 2). To further investigate this genetic interaction, targeted sequencing was used to identify nuclear-encoded complex I variants in an additional 68 AML samples, confirmed by Sanger sequencing (Supplementary Data 3). This combined cohort of 213 patients confirmed a near-complete exclusivity between rare complex I variants and *IDH1* mutation suggesting an epistatic relationship (Fig. 1b, false discovery rate or FDR = $3.5 \times 10^{-08}$, weighted exclusivity test). Notably, *IDH2* somatic mutations did not show significant exclusivity with complex I variants. Furthermore, AML cases with rare complex I variants were likely to have a higher white cell count ($P = 0.06$), higher bone marrow blast percentage ($P < 10^{-6}$), less likely to have French–American–British morphology M2 ($P = 0.003$) and more likely to have a mixed-lineage leukaemia (*MLL/KMT2A*) gene rearrangement ($P = 0.003$) (Supplementary Data 4).

**Mitochondrially encoded complex I genes and *IDH1/2* mutations.** We next used data from an independent AML cohort of 124 de novo cases from Stanford Hospital for which mitochondrial DNA (mtDNA) sequencing data were available[28–31] to investigate the relationship between variants affecting mitochondrially encoded complex I genes and *IDH1/2* somatic driver mutations. This revealed that mtDNA-encoded complex I variants also display near-complete exclusivity with *IDH1* mutations (FDR = $3.26 \times 10^{-5}$, weighted exclusivity test, Fig. 1c), which was not observed for mtDNA variants and *IDH2*. A similar pattern of exclusivity was observed for mtDNA variants in the AML TCGA data[25] (FDR = 0.03, weighted exclusivity test, Supplementary Fig. 1b). To explore this further, we analysed VAF for the complex I variants (nuclear- or mitochondrially encoded) in AML cases across three cohorts where a complex I variant co-occurred with an *IDH1* or *IDH2* somatic mutation. Complex I variants were frequently present at high VAFs (mean VAF = 58.8%) with *IDH2* mutation in line with co-occurrence in the same tumour clone; however, when present in *IDH1*-mutant samples they were at lower VAF (mean VAF = 27.3%; Wilcoxon signed-rank test $P$ value = 0.0088; Fig. 1d). This observation is consistent with complex I variants and the *IDH1* mutation being present in independent clones, or cell-context-specific negative selection of mitochondria[32] and/or mitophagy[33] that maintains the proportion of heteroplasmic mtDNA with the complex I variant below a threshold level. This provides further evidence for incompatibility between rare complex I variants and the oncogenic *IDH1* mutation and suggests increased dependence of *IDH1*-mutant AML on

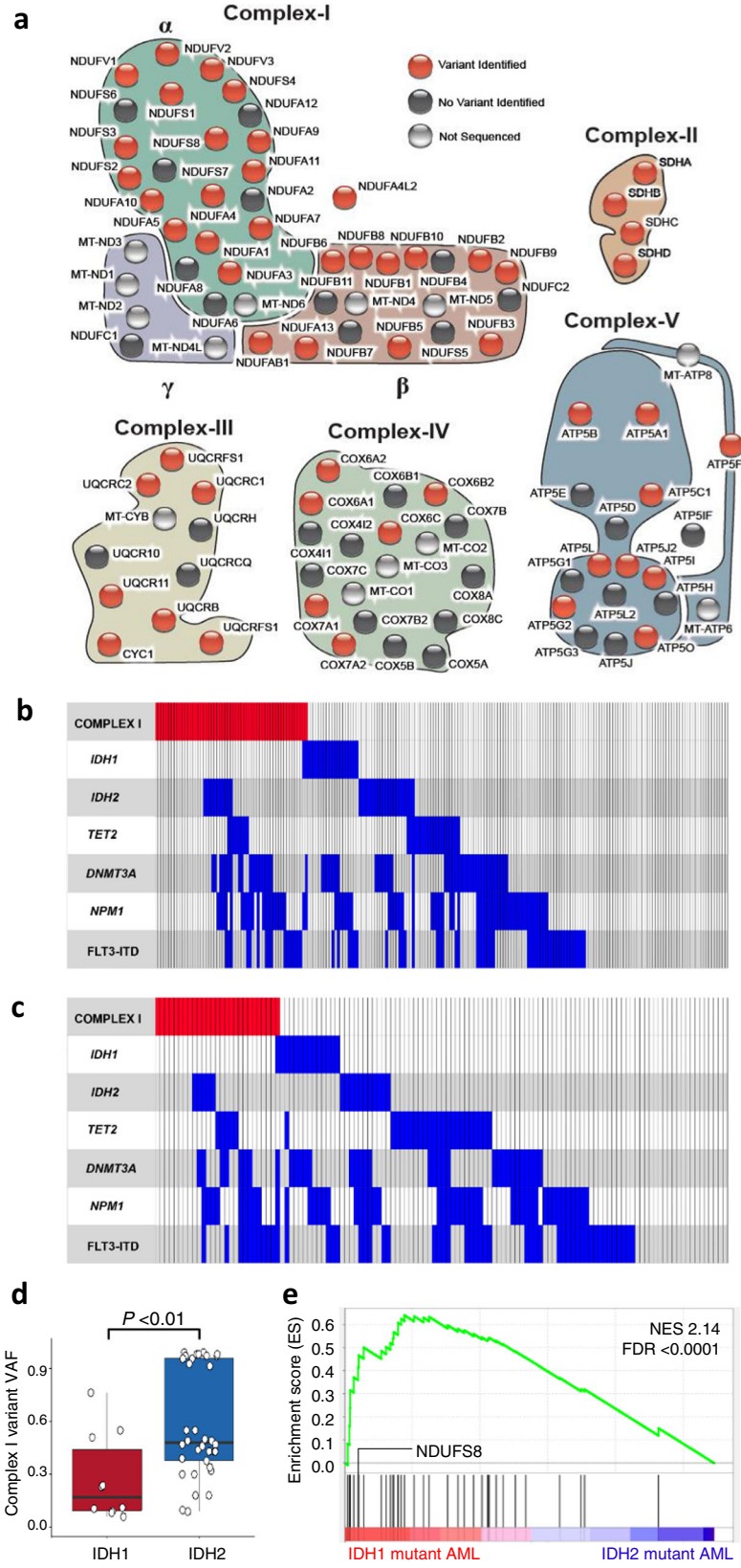

mitochondrial function. Indeed, an increased dependency of *IDH1*-mutant tumours on OXPHOS has been reported[34], and consistent with this, we observed that complex I genes display increased expression in *IDH1*- compared to *IDH2*-mutant AML (Fig. 1e).

**Functional analysis of a rare complex I variant in *NDUFS8*.** Given the unique genetic relationship described above, we next investigated the functional consequences of a rare *NDUFS8* R2C complex I variant that was confirmed heterozygous in the germline of two AML patients, with loss of heterozygosity in the

**Fig. 1 Mitochondrial respiratory chain variants and expression in AML. a** Schematic showing all genes encoding components of respiratory chain complexes I–V. Complex I comprises three subunits α, β and γ. Nuclear-encoded genes with rare variants identified are coloured red. Black indicates nuclear-encoded genes with no variant identified. Grey circles denote mitochondrial-encoded genes; these were not sequenced in the Australian cohort. **b** Co-mutation plot for the Australian patient cohort ($n = 213$). Mutation groups are shown in rows with each individual patient represented by a column. The presence of a mutation is indicated by coloured bar. Near mutual exclusivity was observed between rare nuclear-encoded complex I mutations and somatic *IDH1* R132 mutations (FDR = $3.5 \times 10^{-08}$, weighted exclusivity test). **c** Co-mutation plot for the Stanford patient cohort ($n = 124$) showing segregation of rare mitochondrial-encoded complex I variants and *IDH1* mutations (FDR $3.26 \times 10^{-5}$, weighted exclusivity test). **d** Variant allele frequency (VAF) analysis of complex I variants in patients with co-occurring mutations in either *IDH1* ($n = 10$) or *IDH2* ($n = 34$) in the Australian cohort, Stanford cohort and the Beat AML study[79] ($P = 0.0088$, two-sided Wilcoxon rank sum test). Box and whisker plots indicate median, 25th and 75th percentile and ±1.5 interquartile range. Each dot represents a different sample. Source data are provided as a Source Data file. **e** Positive enrichment of nuclear-encoded complex I genes, including *NDUFS8*, with genes up-regulated in *IDH1*- versus- *IDH2*-mutant AML samples (Beat AML dataset)[79]. Statistical testing performed by gene set enrichment analysis[84], FDR < 0.0001. NES normalized enrichment score.

tumour sample for one such patient (WES-21; Supplementary Fig. 2a). In selected HEK293T clones with confirmed forced expression of wild type (WT) *NDUFS8* (Supplementary Fig. 2b), we did not observe a significant difference in oxygen consumption rate (OCR; a surrogate for respiratory function) relative to parental cells or empty vector controls (Supplementary Fig. 2c). Across three independent experiments, cells expressing the NDUFS8 p.R2C variant exhibited decreased basal and maximal OCR (Supplementary Fig. 2c, d). A trend towards decreased TCA cycle intermediates was also observed in the primary AML sample with NDUFS8 p.R2C variant relative to complex I WT AML samples, with no evidence of increased 2HG (Supplementary Fig. 3). These data suggest that NDUFS8 p.R2C may dominantly suppress mitochondrial respiration. Structurally, *NDUFS8* encodes a highly conserved subunit of the matrix arm of complex I[35]. When visualizing the spatial positioning of NDUFS8 in NADH dehydrogenase utilizing the published *Bos taurus* crystal structure (PDB 5XTH)[35], it was found that the first ~30 amino acids were not captured in the structure provided. Using I-Tasser protein folding prediction[36], we were able to predict that the first ~30 amino acids of NDUFS8 would extend into the matrix arm core of NADH dehydrogenase. This modelling suggests that NDUFS8 Arg2 stabilizes the internal structure of NADH dehydrogenase through scaffolding support to an iron sulfur (Fe–S) cluster (Supplementary Fig. 2e, f and Supplementary Movie 1). For validation of this prediction, we obtained the full-length structure of NDUFS8 as generated by AlphaFold[37] and found that it almost identically matched the prediction by I-Tasser. Thus, the subunit with the p.R2C variant may act in a dominant manner to perturb electron transport when incorporated into complex I. Complex I activity assays in this model system did not show altered activity in cells expressing the NDUFS8 mutant, which may be due to the technical challenges associated with this assay or alternatively may suggest that the impairment in respiratory chain function is more subtle.

**Respiratory capacity of primary AML specimens.** To determine the OXPHOS capacity of primary AML specimens with rare complex I variants, *IDH1*- or *IDH2*-mutant samples, and WT primary AML samples (without complex I variants, *IDH1*- or *IDH2*-mutation), we measured mitochondrial copy number and OCR in bone marrow mononuclear cells from AML specimens (Supplementary Data 5). While the relative mitochondrial copy number was increased in *IDH1*- and *IDH2*-mutant primary AML relative to healthy controls (Fig. 2a) we did not observe significant differences for basal OCR for either *IDH1*- or *IDH2*-mutant AML (Fig. 2b). Metabolite analysis confirmed increased levels of 2HG and reduced levels of TCA intermediates fumarate and malate in the *IDH1*- and *IDH2*-mutant samples (Supplementary Fig. 3). Most notably we observed that maximal OCR, a surrogate for maximal mitochondrial respiratory capacity, determined after

normalizing to basal OCR, was significantly lower for primary AML specimens with complex I variants (*NDUFS8*, *NDUFS3*, *NDUFV2* and *NDUFS1*), and for those samples with *IDH1* and *IDH2* mutations, but not the WT group relative to healthy cells ($P < 0.05$, Fig. 2c, d). These data from primary *IDH1*- and *IDH2*-mutant AML specimens are in agreement with a negative impact of high 2HG levels on OXPHOS capacity[38], the effect of *IDH1* and *IDH2* mutations on the TCA cycle and OXPHOS in AML[39,40], the coupling of the TCA cycle to OXPHOS[3], and the suppressed OCR and reduced TCA cycle activity associated with *IDH1* mutations in glioma[41,42].

**Complex I inhibition as therapy in *IDH1*-mutant AML.** We next directly compared the response of *IDH1*-mutant versus *IDH2*-mutant AML cells to complex I inhibitors utilizing isogenic THP-1 cell line models[38]. Induction of elevated 2HG levels was confirmed for THP1 cells with doxycycline-inducible IDH1 R132H or IDH2 R140Q (Fig. 3a). Inhibition of complex I with rotenone or the more selective, clinical grade complex I inhibitor, IACS-010759[11] resulted in markedly decreased growth over 72 h for cells expressing IDH1 R132H, but not for those expressing IDH2 R140Q (Fig. 3b, c). Combination treatment of parental THP1 cells (IDH1 WT) with IACS-010759 and 2HG over 7 days did not result in synergy nor an additive effect over IACS-010759 treatment alone (Supplementary Fig. 4). We next determined sensitivity to IACS-010759 in a panel of primary AML specimens including *IDH1*-mutant, *IDH2*-mutant, WT AML, and healthy controls (CD34+ cells from healthy donors). Healthy CD34+ cells retained viability following treatment with 5 µM IACS-010759. Viability of the *IDH1*-mutant, but not *IDH2*-mutant primary AML specimens, was consistently suppressed by IACS-010759 relative to *IDH1/2*-WT AML ($P = 0.0033$, Fig. 3d). The differential response observed between *IDH1*- and *IDH2*-mutant AML with complex I inhibition ($P = 0.0396$, Fig. 3d) is consistent with data from solid cancers[34].

The genetic exclusivity and enhanced response of *IDH1*-mutant THP1 cells and primary AML specimens to IACS-010759, compared to *IDH2*-mutant cells, together provide a strong rationale for IACS-010759 as a tailored therapy for the *IDH1*-mutant subtype of AML. Direct oncogene-targeting therapies are now approved by the FDA for both *IDH1*- and *IDH2*-mutant AML[43,44]. Ivosidenib and enasidenib, target the *IDH1*- and *IDH2*-mutant enzymes, respectively, and induce therapeutic responses in AML through relief of epigenetic differentiation block[45,46]. Both therapies are associated with encouraging results in a high percentage of patients although in many cases responses are transient[47], highlighting that alternative treatments are needed to overcome primary and secondary resistance. To investigate complex I inhibition as a potential treatment for these cases, we used lineage markers to determine response to ivosidenib for *IDH1*-mutant AML specimens from the panel in Fig. 3d. Ivosidenib did not induce a differentiation

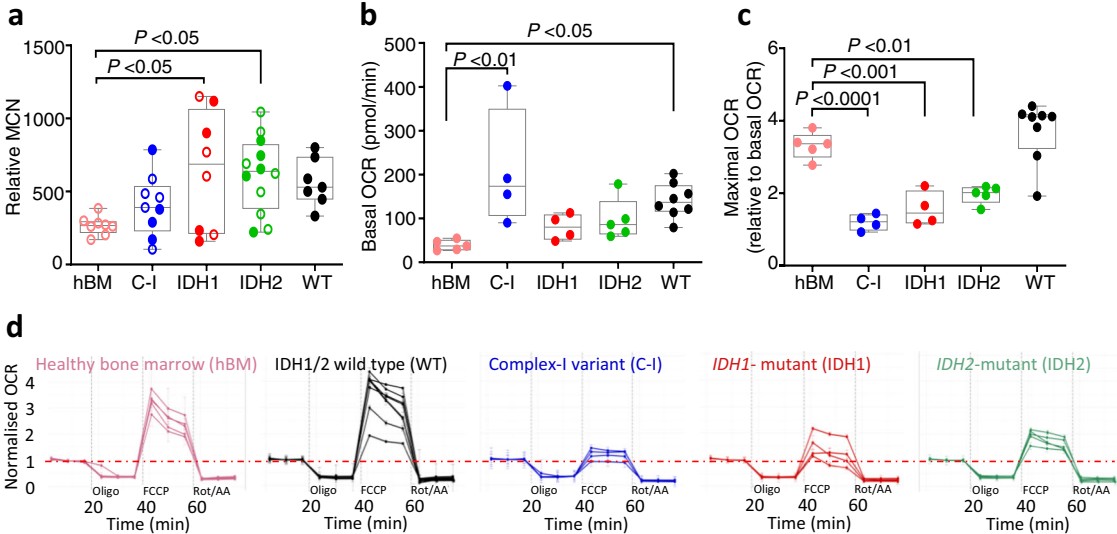

**Fig. 2 Mitochondrial respiration in primary AML samples. a** Relative mitochondrial copy number (MCN) determined by measuring the DNA abundance of mitochondria-encoded cytochrome B (*CYB*) relative to nuclear-encoded glucuronidase beta (*GUSB*) for human bone marrow mononuclear cells from healthy donors (hBM, $n = 8$), complex I (C-I) mutated ($n = 9$), *IDH1*- ($n = 8$), *IDH2*-mutated ($n = 12$), and IDH1/2 wild type (WT, $n = 7$) AML. Samples used to determine oxygen consumption rates in (**b–d**) are shown in filled symbols. Significance determined by one-way ANOVA (Dunnett's multiple correction): $P = 0.016$ (hBM-IDH1), 0.013 (hBM-IDH2). **b** Basal oxygen consumption rate (OCR) and (**c**) maximal OCR normalized to basal OCR as measured after addition of uncoupling agent for hBM ($n = 5$), C-I mutated ($n = 4$), *IDH1*- ($n = 4$), *IDH2*-mutated ($n = 5$) and WT ($n = 8$) AML samples. For (**b**) and (**c**), significance determined by one-way ANOVA (Dunnett's multiple correction): significance values for (**b**): $P = 0.0015$ (hBM vs. C-I) and 0.026 (hBM vs. WT); significance values for (**c**): $P < 0.0001$ (hBM vs. C-I), $P = 0.0005$ (hBM vs. IDH1) and 0.0035 (hBM vs. IDH2). Box and whisker plots indicate median, 25th and 75th percentile and range of data. Each dot indicates a different patient sample. **d** Normalized OCR for the individual samples summarized in (**b**) and (**c**) showing reduced OXPHOS capacity in AML with rare complex I variants ($n = 4$) and *IDH1*- ($n = 4$) and *IDH2*- mutated ($n = 5$) AML samples, compared to wild type (WT) samples ($n = 8$) and hBM controls ($n = 5$). WT samples are FLT3-ITD negative and WT for *IDH1, IDH2, DNMT3A, NPM1*, and complex-I. Each line represents an independent patient sample tested in triplicate and each value is the mean of a given measurement timepoint. Error bars represent SD. Data represented as fold change relative to basal OCR. Details of patient samples are provided in Supplementary Data 5. Source data are provided as a Source Data file.

response in three of the five samples tested, consistent with primary resistance (Supplementary Fig. 5). Irrespective of ivosidenib response, all five *IDH1*-mutant primary AML specimens displayed sensitivity to IACS-010759 (Fig. 3d). Thus, IACS-010759-based therapies could be considered for treatment of AML cases that are resistant to ivosidenib; supporting this, administering ivosidenib and IACS-010759 concurrently in a panel of *IDH1*-mutant AML patient-derived xenografts (Pdx) has recently been reported to improve responses, relative to ivosidenib alone, for some *IDH1*-mutant samples[40]. The heterogenous response may not be surprising given that co-treatment with ivosidenib may reverse (at least partially) the *IDH1*-mutant phenotype; indeed, we show that treatment with ivosidenib generates partial or full resistance to IACS-010759 in two of five *IDH1*-mutant primary samples (Fig. 3e). These studies highlight that the benefit of combination treatment with ivosidenib and complex I inhibitors is likely to be highly patient-specific. More studies will be needed to define the metabolic, epigenetic and gene expression changes that are impacted by ivosidenib across *IDH1*-mutant AML samples.

**Metabolic vulnerability of *IDH1*-mutant AML.** To determine the mechanism of differential sensitivity of *IDH1*-mutant AML to IACS-010759, we investigated the ATP dynamics of *IDH1*-mutant versus *IDH2*-mutant AML cells. Simultaneous measurement of ATP generation via OXPHOS or glycolysis in the isogenic THP-1 cell line models showed lower total ATP in both *IDH1*- and *IDH2*-mutant cells compared to their respective wild type controls (Fig. 4a, Supplementary Fig. 6a, b). While increased basal mitochondrial ATP as a percentage of total has been reported for a combined group of *IDH1/2*-mutant primary AML samples[40] we

observed this only for *IDH2*-mutant THP1 cells (Fig. 4b). Glycolytic ATP production was increased following IACS-010759 treatment (versus basal) in all cells (Fig. 4a). However, with IACS-010759, the glycolytic ATP production of *IDH1*-mutant THP1 cells was significantly less than IDH1-WT controls; in contrast, the *IDH2*-mutant THP1 cells had higher glycolytic capacity compared to IDH2-WT cells (Fig. 4c). *IDH2*-mutant primary samples also showed increased glycolytic spare reserve relative to *IDH1/2* WT samples (Supplementary Fig. 6c, d). For a panel of primary CD34+ and AML samples we observed a consistent switch to glycolysis as the major source of ATP when treated with IACS-010759, irrespective of *IDH* mutation status, although this was associated with a high degree of variability in the proportion of glycolysis/OXPHOS observed across untreated samples (Fig. 4d, Supplementary Fig. 7). Thus, despite some variability, both *IDH1*- and *IDH2*-mutant AML cells maintain the capacity to switch to glycolysis for ATP production when challenged with complex I inhibition. Given that the IDH1 reaction plays a non-redundant role as a critical source of cytosolic NADPH[15,48], we also considered that a selective deficiency in ability of *IDH1*-mutant AML to maintain NADPH levels may contribute to the differential sensitivity to complex I inhibition. Importantly, in *IDH1*- and *IDH2*-mutant AML lethal levels of ROS could arise due to rapid consumption of NADPH by the oncogenic IDH mutant enzymes[49–51]. In primary patient samples NADPH levels were significantly reduced relative to CD34+ controls for *IDH1*-mutant, but not *IDH2*-mutant AML (Fig. 4e), consistent with reports from studies with *IDH1*-mutant solid tumours[41,49,52].

We propose that as glutamine can be utilized via *IDH1*-driven reductive carboxylation[53], in IDH1 WT AML, flexibility of

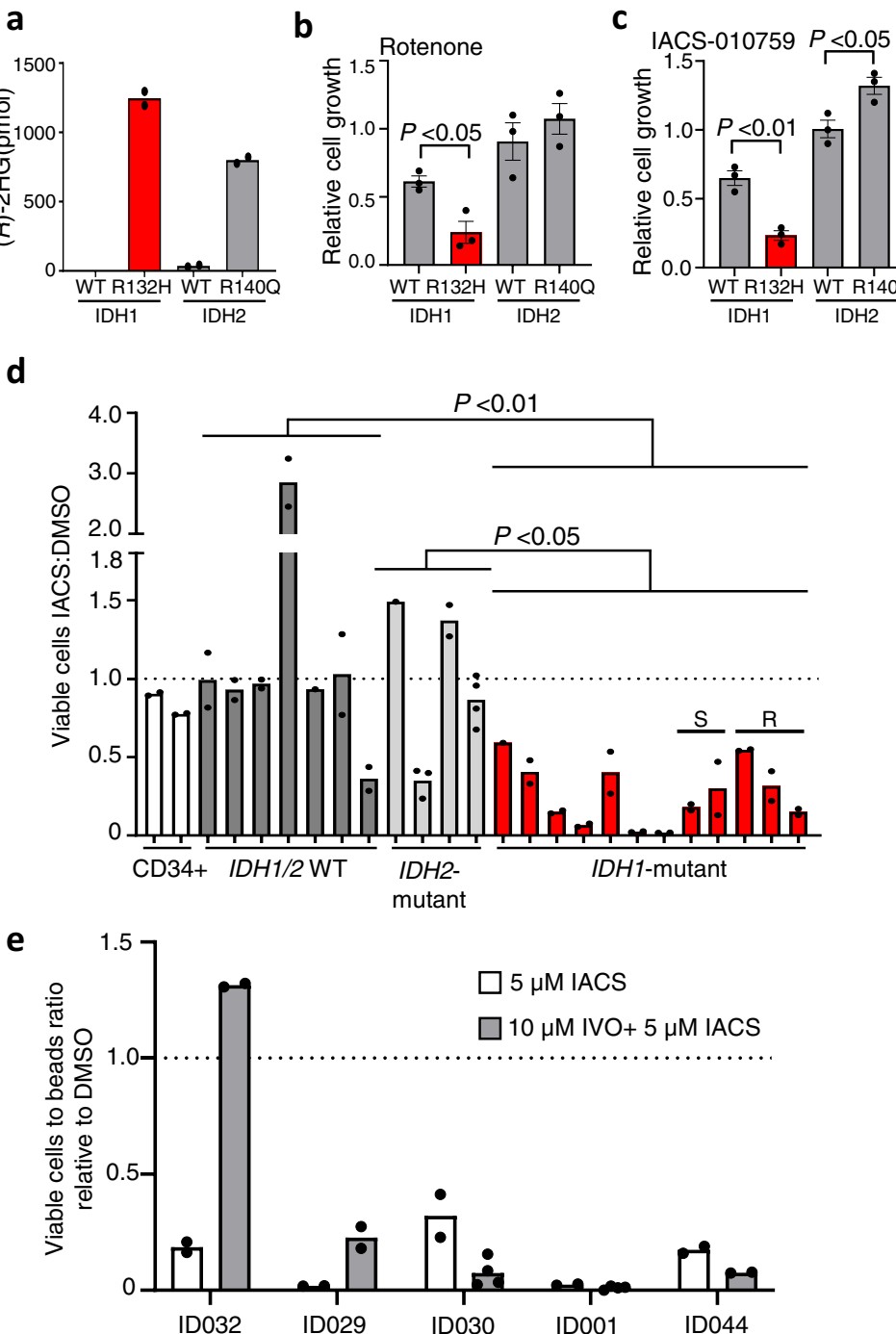

**Fig. 3 _IDH1_ mutation confers sensitivity to complex I inhibition. a** (R)-2-hydroxyglutarate (2HG) abundance in supernatants from THP-1 cells transduced with doxycycline-inducible IDH1-wild type (WT), IDH1 R132H, IDH2-WT, or IDH2 R140Q. Bars represent mean of technical duplicates from a single experiment. Data points for _IDH1_-WT were below background and thus, are not represented in the graph. **b** and **c** Cell growth of GFP+ doxycycline-inducible _IDH1_-WT, _IDH1_ R132H, _IDH2_-WT or _IDH2_ R140Q expressing THP1 cells at 72 h after treatment with (**b**) 1 μM rotenone or (**c**) 5 μM IACS-010759, relative to DMSO vehicle control. Bars represent mean ± SEM from three independent experiments. Each dot represents the mean of an independent experiment. Statistical significance determined by two-tailed unpaired t-test. Significance values for (**b**) P = 0.0146 (IDH1-WT vs. IDH1 R132H), and (**c**) P = 0.0028 (IDH1-WT vs. IDH1 R132H) and 0.0246 (IDH2-WT vs. IDH2 R140Q). **d** Ratio of live (propidium iodide-negative) cells for healthy CD34+ samples (n = 2), _IDH1/2_ WT (n = 7), _IDH2-_ (n = 4) or _IDH1_-mutant (n = 12) AML samples treated with 5 μM IACS-010759 (IACS), relative to DMSO vehicle control over 72 h. Samples tested for response to ivosidenib in Supplementary Fig. 5 are shown as S, sensitive or R, resistant to ivosidenib. Values represent mean for each sample in a single experiment and each dot is a data point. Significance determined for patient groups using one-way ANOVA (Tukey's multiple correction), P = 0.0033 (IDH1/2 WT vs. IDH1 mutant), 0.0396 (IDH1 mutant vs. IDH2 mutant). **e** Variable response of IDH1-mutant samples treated for 72 h with 5 μM IACS-010759 (IACS) following 24 h pre-treatment with 10 μM ivosidenib (IVO) compared to IACS alone. Data presented relative to DMSO control. Data show mean of a single experiment and each dot represents a data point. Details of patient samples used in d-e are provided in Supplementary Data 5. Source data are provided as a Source Data file.

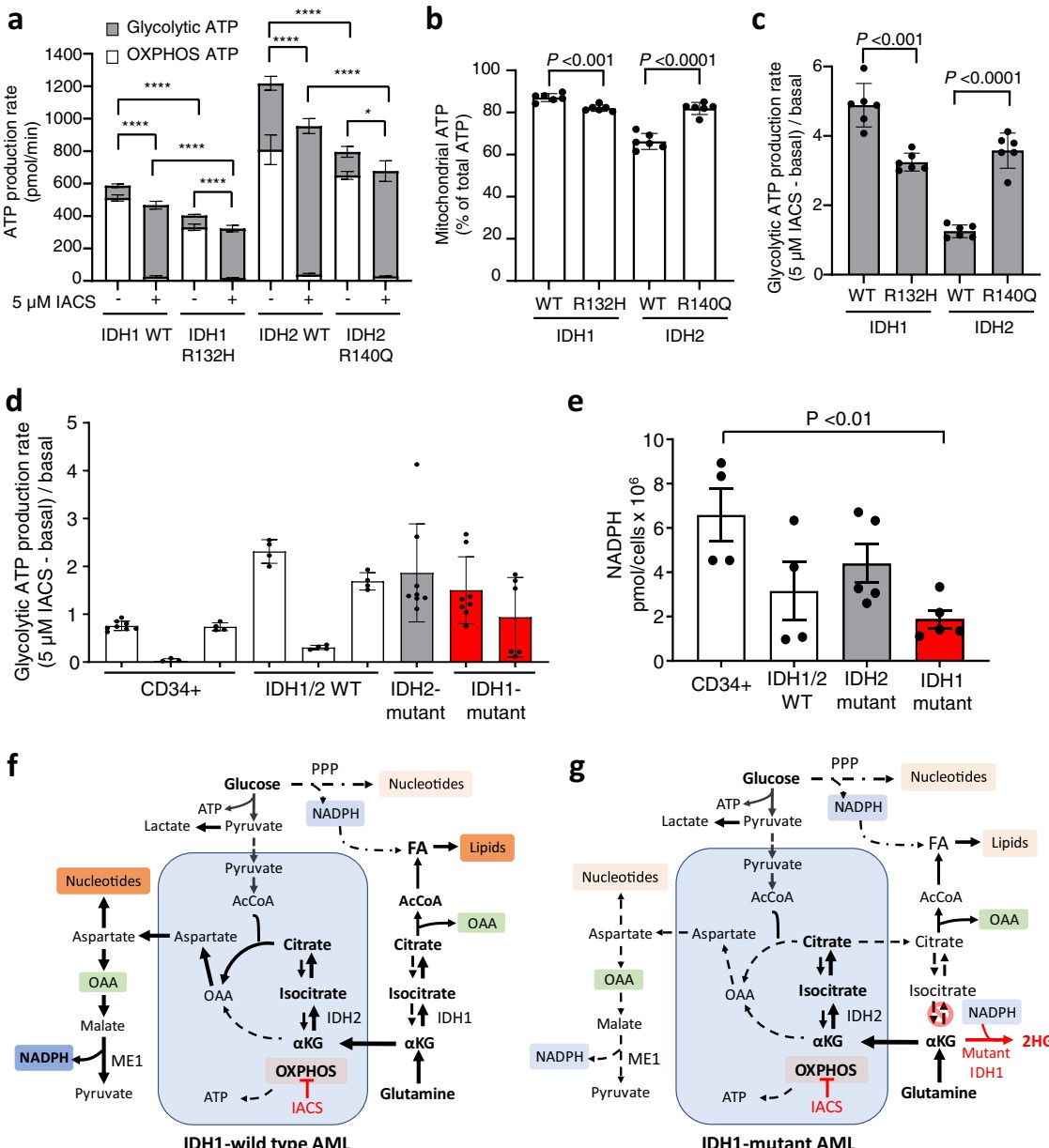

**Fig. 4 Metabolic plasticity of *IDH1*-mutant AML. a** Total ATP production rate at basal and following induction with 5 μM IACS-010759 (IACS) for isogenic THP1 cells expressing IDH1 wildtype (WT), IDH1 R132H, IDH2-WT or IDH2 R140Q, measured by ATP rate assay. Contribution to ATP production rate from glycolysis (grey) and OXPHOS (white) are shown as mean ± SD of six technical replicates. Statistical significance determined by one-way ANOVA (Tukey's multiple correction), ****$P < 0.0001$ and *$P = 0.029$. **b** Mitochondrial ATP as a percentage of total ATP in untreated isogenic THP1 cells shown as mean ± SD of six technical replicates. Significance determined by two-tailed unpaired $t$-test, $P = 0.0005$ (IDH1-WT vs. IDH1 R132H) and $P < 0.0001$ (IDH2-WT vs. IDH2 R140Q). **c** Change in glycolytic ATP following addition of 5 μM IACS-10759 (IACS) relative to basal glycolytic ATP production rate, measured by ATP rate assay for isogenic THP1 cells. Data shown as mean ± SD of six technical replicates. Significance determined by two-tailed unpaired $t$-test, $P = 0.0001$ (IDH1-WT vs. IDH1 R132H) and $P < 0.0001$ (IDH2-WT vs. IDH2 R140Q). Raw oxygen consumption rate (OCR) and extracellular acidification rate (ECAR) data related to (**a**–**c**) are presented in Supplementary Fig. 6a, b. **d** Change in glycolytic ATP following addition of 5 μM IACS-10759 relative to the basal glycolytic ATP production rate, for healthy CD34+ samples ($n = 3$), *IDH1/2* WT ($n = 3$), *IDH2*- ($n = 1$) or *IDH1*-mutant ($n = 2$) AML. Data presented as mean ± SD of four or more technical replicates per sample, no significance was observed between groups. Individual sample results in Supplementary Fig. 7. **e** NADPH levels in CD34+ samples ($n = 4$), *IDH1/2* WT ($n = 4$), *IDH2*- ($n = 5$) or *IDH1*-mutant ($n = 5$) AML. Data presented as mean of the group ± SEM. Each dot is the mean of an independent sample. Significance determined by one-way ANOVA (Dunnett's multiple correction), significance $P = 0.0091$ (CD34+ vs. IDH1 mutant). Details of patient samples used in (**d**) and (**e**) are provided in Supplementary Data 5. Source data for (**a**–**e**) are provided in Source Data file. Schematic representation of metabolic rewiring of (**f**) *IDH1* wild type AML and (**g**) *IDH1* mutant AML following complex I inhibition. See text for explanation. Mitochondria shaded in blue. AcCoA acetyl CoA, FA fatty acid, IACS IACS-010759, IDH isocitrate dehydrogenase, ME1 malic enzyme 1, OAA oxaloacetic acid, OXPHOS oxidative phosphorylation, PPP pentose phosphate pathway, αKG alpha ketoglutarate, 2HG 2-hydroxyglutarate.

glutamine utilization provides an adaptive survival mechanism when complex I and the TCA cycle are impaired or inhibited; this promotes mitochondrial citrate usage for the non-canonical pathway for aspartate (via citrate synthase) to support nucleotide synthesis and NADPH maintenance (Fig. 4f)[15,54–56]. Furthermore, we propose that in *IDH1*-mutant cells, in which neomorphic IDH1 enzyme rapidly consumes cytosolic NADPH, the impairment of OXPHOS/TCA cycle due to rare germline complex I variants or pharmacological inhibition, results in a critical demand on mitochondrial citrate (generated via IDH2) and aspartate (Fig. 4g). Indeed, treatment of *IDH1*-mutant AML with IACS-010759 has been reported to suppress aspartate levels[40], consistent with a high demand for mitochondrial citrate that is required to support citrate levels in the cytosol for acetyl CoA production, fatty acid biosynthesis, acetylation of histones and other proteins[57].

## Discussion

While germline mutations can modify the effects of somatic driver mutations in cancer[58], to our knowledge the epistatic relationship that we report here between germline variants and somatic oncogenic mutations has not been described previously in cancer. The germline complex I variants appear to have a subtle effect on their own, reducing maximal respiratory capacity in line with the *NDUF11* knockdown phenotype reported in *Caenorhabditis elegans*[59]. We propose that the reduced capacity to boost OXPHOS in the presence of a rare germline complex I variant precludes the proliferation and survival of cells under hematopoietic stress if they acquire a somatic *IDH1* mutation. This may be due in part to compensatory utilization of complex II for OXPHOS[59] which will generate detrimental levels of ROS in the absence of sufficient NADPH. Importantly, a sub-clinical red blood cell phenotype has recently been associated with inherited mitochondrial-encoded variants in complex I genes[60], suggesting the possibility that rare germline complex I variants confer a hematopoietic phenotype that includes inherent resistance to acquisition of the IDH1 oncogenic mutation. Our identification of this genetic interaction between complex I gene variants and *IDH1* mutation has highlighted the potential for similar exclusivities and unique metabolic vulnerabilities in other cancer settings. Thus, inclusion of germline variants that are commonly filtered out in cancer studies will be important for future analysis of cancer cohorts.

The unique epistatic relationship between complex I variants and oncogenic IDH1 mutation suggested a specific AML context where a unique metabolic vulnerability is targetable. IDH1 driver mutations can arise early in leukemogenesis in pre-leukaemic clonal hematopoietic stem cells (HSC) populations[27] that persist following chemotherapy providing a source of post-treatment relapse[61]. The unique genetic and pharmacological vulnerability to complex I impairment described here for *IDH1*-mutant AML provides a strong rationale for the incorporation of complex I inhibitor therapy in the treatment of *IDH1*-mutant AML. This is further supported by responses observed in some Pdx models of *IDH1*-mutant AML to IACS-010759[40] and reports that wild type HSC have low OXPHOS levels and are insensitive to complex I inhibition[10,11,14]. We and others[16,62] report that the majority of AML subtypes are likely to have the metabolic plasticity to adapt to OXPHOS inhibition via metabolic rewiring (e.g. utilization of glycolysis and/or glutaminolysis). It will now be important to carefully assess the sensitivity of a large panel of AML to IACS-010759 treatment. A wide range of sensitivity has been reported for AML samples with other complex I inhibitors[10], and metabolic profiles and vulnerabilities may vary considerably across subtypes; for example the non-quiescent, cycling leukaemic stem cells in

MLL-rearranged leukaemias, and AML over-expressing TET3, may show increased dependence on glucose metabolism[63,64].

IACS-010759 has a unique mechanism of action making it clinically favourable[12,65]. It shows promise in preclinical studies of aggressive solid cancers, and AML[11,40,66–68], and is in clinical trial[14]. In addition to IACS-010759, there are other clinically available complex I inhibitors, e.g., mubritinib[10] and CPI-613[14] that may provide important alternatives to IACS-010759. The anti-diabetic drug metformin, which has an excellent safety profile and is in a clinical trial for *IDH1* mutant solid tumours[69], inhibits complex I and may be suitable for maintenance therapy aimed at targeting residual preleukemic clones in *IDH1*-mutant AML patients in remission. Response of AML cells to complex I inhibition in vivo may be affected by several adaptive mechanisms including transfer of mitochondria from bone marrow stromal cells[70,71], extracellular uptake of citrate[57] or aspartate[9], increased utilization of fatty acid oxidation[40,72]. Thus adjunctive therapies may be required to optimize clinical response to complex I inhibitors; these could include the manipulation of citrate levels in the AML bone marrow niche[57], or inhibition of fatty acid uptake[73].

For *IDH1*-mutant AML encouraging results have been reported using ivosidenib, a selective targeted inhibitor of the mutant oncogenic IDH1 enzyme. Around 40% of *IDH1*-mutant patients with relapsed/refractory AML that receive this inhibitor achieve complete remission with a median survival of around 1 year[27,43,74,75]. However, alternative treatments are needed in the setting of ivosidenib resistance. Given that the most likely clinical scenario will involve patient treatment with ivosidenib until relapse, either as monotherapy or combined with chemotherapy, it is critical to consider the most appropriate strategy for targeting OXPHOS in ivosidenib-resistant *IDH1*-mutant AML cases. We propose that sequential treatment with ivosidenib followed by IACS-010759 once patients relapse, rather than concurrent administration, is the most rational clinical approach and is in line with recent studies that highlight the importance of investigating sequential targeting strategies for *IDH*-mutant AML[74]. This approach also considers our data showing that at least in some patients, effective inhibition of mutant IDH1 antagonizes the activity of the complex I inhibitor. Such reversal of the effect of complex I inhibition following treatment with mutant IDH1 inhibitor is consistent with the critical role of reductive carboxylation in the adaptive response to IACS-010759 in *IDH1*-mutant tumours[52,53], and increased levels of citrate and other TCA cycle intermediates following treatment of *IDH1*-mutant AML cell lines;[40] such reversal of the *IDH1*-mutant metabolism may explain the response of only some Pdx models to combination therapy using IACS-010759[40].

In summary, we show an epistatic interaction between rare complex I germline variants and somatically acquired *IDH1* oncogenic mutation that can be explained by the role of IDH1 in supporting anabolic demands and compensatory NADPH production that allows rescue from impaired complex I activity. We propose that the lack of this metabolic plasticity in *IDH1*-mutant AML confers hyper-sensitivity to pharmacological inhibition of complex I and demonstrate that inhibition of complex I is a promising strategy for *IDH1*-mutant AML patients with primary ivosidenib resistance.

## Methods

**Patient samples**. AML samples were collected from patients and volunteers with written, informed consent for use of their samples in research by the South Australian Cancer Research Biobank (Adelaide, SA, Australia) and the tissue bank of the Australasian Leukaemia & Lymphoma Group (Melbourne, VIC, Australia). Clinical characteristics for the discovery cohort are shown in Supplementary Data 2. AML patient samples were donated at the time of routine diagnostic and follow-up assessments and no compensation was paid. Volunteer healthy donors

were paid an honorarium for their participation. This study was approved by the ethics committees of the Royal Adelaide Hospital (Ethics Approval HREC/12/RAH/152 and HREC/18/CALHN/588), the University of South Australia (Approval Number 0000031791), the Princess Alexandra/Metro South Health (Ethics Approval HREC/04/QPAH/172) and the University of Queensland (Approval Number 2013001109). For Stanford Hospital AML samples primary peripheral blood and bone marrow samples were obtained from patients with de novo AML prior to treatment, with informed consent according to institutional guidelines (Stanford University Institutional Review Board No. 6453). No compensation was paid. All primary AML samples used in functional assays are listed in Supplementary Data 5.

**Sequencing analyses.** Whole exome sequencing of the Australian AML cohort has been previously reported[22]. Rare complex I variants were identified as those with a minor allele frequency < 0.005 as reported in dbSNP137, 1000 Genomes (April 2012, any ethnicity) or the National Heart, Lung, and Blood Institute Exome Sequencing Project (evs-6500, any ethnicity). For Ion-Torrent sequencing, primers were designed for a custom amplicon panel using Ion Torrent's AmpliSeq Designer (version 4.24) targeting the nuclear-encoded complex I genes. All primer sequences are listed in Supplementary Data 6. Libraries were sequenced on the Ion-Torrent Proton P1 chip aiming for >1000 × mean coverage. Sequences were aligned to the hg19 reference genome using Torrent Mapping Alignment Programme and variants were called as part of Torrent Suite v4.0 using stringent settings. Variants were filtered based on design content, read quality and strand bias. Filtered variants were annotated using an in-house pipeline. Variants were confirmed by Sanger sequencing. For all variants identified in the Australian cohort, population variant frequencies from the latest version of gnomAD are provided in Supplementary Data 1 and 3. For Stanford Hospital samples, exome sequencing has been previously reported[28–31].

**Preparation of AML samples for metabolic studies.** Lymphoid cells were depleted from bone marrow and blood samples using magnetic separation. Briefly, $1 \times 10^7$ cells were suspended in cold rinsing solution (Miltenyi Biotec, 130-091-222) supplemented with bovine serum albumin and incubated with anti-CD3 and anti-CD19 MicroBeads (Miltenyi Biotec, 130-050-101 and 130-050301, respectively) before passing through filters and columns. To remove dead cells, up to $1 \times 10^7$ cells were suspended in Dead Cell Removal MicroBeads (Miltenyi Biotec, 130-090-101) and filtered through magnetic columns. Cells were recovered in primary media with cytokines for 2 h at 37 °C prior to performing experiments.

**Measurement of oxygen consumption rate and glycolysis.** Lymphoid-depleted patient samples were suspended in XF Assay media (Agilent Seahorse Bioscience, 102353-100) with pH adjusted to 7.4 ± 0.1 supplemented with 4.5 g/l glucose (Sigma-Aldrich, G8769), 0.11 g/L sodium pyruvate (Sigma-Aldrich, P5280) and 8 mM L-glutamine (Sigma-Aldrich, G-7513). 220K cells were added to each well of XFe or XFp Cell-Tak (Corning, 354240) pre-coated culture plates and then slowly centrifuged for incubation at 37 °C non-CO₂ incubator. Oxygen consumption rate was measured at baseline using either Seahorse XFe96 or XFp analyzers according to standard protocols and after the addition of oligomycin (2 μM), carbonyl cyanide-4-(trifluoromethoxy) phenylhydrazone (2 μM) and rotenone and antimycin A (0.5 μM). Fold change was determined by normalizing raw values to the average of the second basal reading. Extracellular acidification rates were measured using the Glyco Stress Test™ collected at baseline and after the addition of 10 mM glucose, oligomycin (final concentration in well 2 μM) and 2-deoxy-glucose (50 mM). Fold change was determined by normalizing raw values to the average of the second basal reading.

**Metabolite analysis.** Metabolite extraction was performed on lymphocyte-depleted bone marrow samples ($0.85–1 \times 10^6$ cells). Cell pellets were washed in phosphate-buffered saline and snap frozen in ethanol:dry-ice and stored at −80 °C. Samples were extracted using chloroform:methanol:water ($CHCl_3:CH_3OH:H_2O$, 1:3:1 v/v/v) with the supernatant then adjusted by the addition of $H_2O$ to a final ratio ($CHCl_3:CH_3OH:H_2O$, 1:3:3 v/v/v) to induce phase partitioning essentially as described in Masukagami et al. [76]. Analysis was performed by Metabolomics Australia (Melbourne) for selected sugars and organic acids using GC–QqQ–MS[77], and for 2-hydroxyglutaric acid using GC–MS[78]. Data was acquired in MRM mode. Absolute concentrations (μM) of targeted sugars and organic acids were quantified using linear response of the corresponding calibration series of authentic standards. GC–QqQ–MS data were processed using the Agilent MassHunter Quantitative Analysis version B.07.00 software. Mass spectra of eluting TMS compounds were identified and quantified using corresponding calibration series of authentic standards. Results were normalized to internal standard, $^{13}C_6$-Sorbitol and number of cells, and expressed as picomoles/cell.

**Quantitation of mitochondria number.** Relative mitochondria copy number was calculated based on quantifying the amount of mitochondrially encoded cytochrome B (*MT-CYB*, Life Technologies inventoried assay: Hs02596867-s1,FAM-MGB), normalized to that of nuclear-encoded glucuronidase beta (*GUSB*, Life Technologies custom probe) by quantitative real-time PCR. Reactions were set up

using TaqMan® Universal PCR Master Mix (Applied Biosystems). Amplification of *GUSB* was performed using forward (ATTTTGCCGATTTCATGACTGA) and reverse (GACGGGTACGTTATCCCATGAG) primers, with custom MGB-probe (AGTGTAAGTGGCAGTTTG) using 20 ng of genomic DNA. *MT-CYB* was amplified using concentrations as specified by the manufacturer using the Applied Biosystems ViiA™ 7 Real-Time PCR System and QuantStudio Real-Time PCR Software v1.1 (Life Technologies). Relative copy number per diploid cells was calculated as follows: Mito copy number = $2 \times (2-\Delta Ct)$, where $\Delta Ct = (Ct\ MT-CYB$-Ct $GUSB)$.

**Culture of primary CD34+ cells and leukaemic blasts.** AML blasts (FACS-sorted; low side scatter, CD33⁺, CD45^mid) were cultured in IMDM (Sigma-Aldrich, I3390) with 20% foetal calf serum (CellSera, AU-FBS/PG), 50 ng/ml human stem cell factor (Peprotech, 300-07) and 10 ng/ml each of thrombopoeitin (Peprotech, 300-18), FLT-3 ligand (Peprotech, 300-19), interleukin-3 (Peprotech, 200-03), interleukin-6 (Peprotech, 200-06), and granulocyte-colony stimulatory factor (Peprotech, 300-23) as well as 100 μM β-mercaptoethanol (BME, Sigma Aldrich, M3148). For drug treatment experiments serum was added to 0.5% and BME was excluded. CD34+ cells were isolated using a human CD34 MicroBead Kit (Miltenyi Biotec, 130-046-702). CD34+ cells were cultured in IMDM with 20% foetal calf serum and 20 ng/ml interleukin-6, and 100 ng/ml each of stem cell factor, FLT-3 ligand and thrombopoietin, 35 nM UM171 (Stem Cell Technologies, 72332) and 0.75 μM StemRegenin1 (Stemcell Technologies, 72344). IACS-010759 was obtained from SelleckChem (S8731).

**Ivosidenib treatment and differentiation of primary AML specimens.** Lymphoid-depleted patient AML samples were cultured in IMDM supplemented with 20% foetal calf serum, 50 ng/ml human stem cell factor and 10 ng/ml each of thrombopoietin, FLT-3 ligand, interleukin-3, interleukin-6, and granulocyte-colony stimulatory factor (details listed above), 100 μM β-mercaptoethanol (BME, Sigma Aldrich, M3148) together with 10 μM ivosidenib (SelleckChem, S8206) or DMSO (control) for up to 8 days. Differentiation of cells was measured using CD33-APC, CD11b-PE, CD11c-BV421, CD14-PerCP-Cy5.5, CD15-FITC, and CD16-PE-Cy5 (BD Biosciences) in viable cells (determined by forward and side scatter profiles) on a BD FACSCanto™II flow cytometry system. Differential expression for each marker was determined by FloJo™.

**Quantification of ATP production.** Lymphoid-depleted patient AML samples were suspended in XF RPMI Assay supplemented with 10 mM glucose, 1 mM pyruvate and 2 mM glutamine (Agilent Seahorse Bioscience, 103681-100), with pH adjusted to 7.4 ± 0.1. $1.2 \times 10^5$ cells were added to each well of XF96 culture plates pre-coated with Cell-Tak (Corning, 354240) and incubated at 37 °C in a non-CO₂ incubator. Oxygen consumption rate was measured at baseline using either Seahorse XFe96 analyzer according to standard protocols and after the addition of oligomycin (1.5 μM) and rotenone and antimycin A (0.5 μM). IACS-010759 was added in Port A (5 μM final concentration in well) as per XF real-time ATP rate assay (Induced) Kit (Agilent Technologies, 103592-100) and analysed with Wave software (Agilent Technologies, San Jose, CA). For simultaneous measurement of ATP generated from glycolysis and OXPHOS, we used an established protocol based on measured ratios of the moles of ATP generated per mole of oxygen consumed, assuming complete oxidation of glucose yields up to 33.45 ATP.

**Quantification of NADPH.** Prior to measurement, primary AML blast cells were cultured for 48 h following thawing and assessed for viability of >75%. Five million live cells were used to measure NADPH using the High Sensitivity NADPH Quantitation Fluorometric Assay Kit (Sigma Aldrich, MAK216) following the manufacturer's protocol.

**Cell lines.** THP-1 cells and HEK293T cells were sourced from ATCC (TIB-202 and CRL-3216, respectively).

**Complex I inhibitor assays using doxycycline-inducible IDH1 mutant cells.** IDH1-WT, IDH1-R132H, IDH2-WT, and IDH2-R140Q proteins were expressed in THP-1 cells using pTRIPZ (Open Biosystems) tet-inducible lentiviral vector with green fluorescent protein encoded in the same open-reading frame by T2A peptide[38]. After 4 days of doxycycline induction cells were plated at $10^5$ cells per ml in RPMI supplemented with 0.5% serum, with complex I inhibitors. Rotenone and doxycycline were obtained from Sigma.

**Quantitation of (R)-2-hydroxyglutarate.** (R)-2-hydroxyglutarate was quantitated in cell culture media after 4 days of doxycycline induction using the enzyme (D)-2-hydroxyglutarate dehydrogenase coupled to reduction of NAD+ (Sigma-Aldrich. MAK320). Briefly, cell culture supernatant was de-proteinised (De-proteinising sample preparation kit, Biovision, K823-200) with perchloric acid solution and neutralized with potassium hydroxide. Standards were made fresh before incubation at 37 °C for 60 min and plate read at $\lambda_{em}$ = 590 nM.

**Combination treatment of THP1 cells**. Parental THP1 cells seeded at $5 \times 10^4$ cells per ml were cultured for 7 days in RPMI supplemented with 1% serum in the presence of either 300 µM (2R)-Octyl-2-hydroxyglutarate (Med Chem Express, HY-103641), 5 µM IACS-010759 (SelleckChem, S8731), the combination of both or DMSO control. The ratio of live DAPI-positive cells to CountBright™ Absolute Counting Beads (ThermoFisher Scientific, C36950) were determined on days 3, 5, and 7 using the BD FACSCanto™II flow cytometry system.

**Protein folding prediction**. The cryo-electron microscopy structure of the supercomplex $I_1III_2IV_1$ (PDB 5XTH)[35] was used for positional localization of NDUFS8 with relation to the remainder of the NADH dehydrogenase structure. For structural prediction of full-length NDUFS8, I-Tasser[36] was utilized, while a precomputed structure of NDUFS8 was available directly from AlphaFold (https://alphafold.ebi.ac.uk/)[37]. All proteins were visualized and rendered using PyMol v2.1.0.

**Gene set enrichment analysis**. Gene expression measurements in tabular count format for 451 specimens from 411 patients was retrieved from the Beat AML Functional Genomic Study[79]. Patient data were stratified into two groups based on *IDH* mutation state (*IDH1*:36 and *IDH2*:42 patients). Differential expression analysis between the two groups was evaluated from TMM normalized gene counts using R (version 3.6.3) and edgeR (version 3.3)[80] following protocols as described[81]. Only genes with a count per million (CPM) > 3 in more samples than the smallest group being compared were retained for further analysis. Alignments were visualized and interrogated using the Integrative Genomics Viewer v2.3.80[82]. Graphical representations of the results were generated using Glimma[83]. The Gene Set Enrichment Analysis software package (GSEA v4.1.0)[84] was used to look for coordinate expression against custom built gene sets. Genes were ranked for GSEA analysis (GSEAPreranked) by calculating the "directional" negative log FDR (sign of fold change * −log10(FDR)).

**Statistical analysis**. Data processing and statistical analysis was performed in R. Weighted exact test for mutual exclusivity was performed as described by ref. [85]. Statistical analysis of the metabolite measurements data and pharmacological treatments was performed using Graphpad Prism v9.

**Reporting summary**. Further information on research design is available in the Nature Research Reporting Summary linked to this article.

## Data availability

The relevant data supporting the key findings of this study are available within the article and its Supplementary Information and Data files or from the corresponding author upon reasonable request. Source data are provided with this paper. The variant data used in this study were obtained from the WES described previously[22] and are available in the European Genome–Phenome Archive (EGA) database with the accession code # EGAS00001006185. Data access is restricted in accordance with conditions provided for publication of data in the patient informed consent sheet, approved by the Institution Human Research Ethics Committee. These restrictions are to maintain patient privacy, avoid patient identification and maintain a record of who has access to the data. Application for access to the data can be done directly through the EGA website above. There are no restrictions on who may access the data provided the conditions for access listed on the website (https://ega-archive.org/studies/EGAS00001006185) and detailed in the Data Access Agreement (available during the data request process) are met. Data will be accessible for an unlimited period once the relevant Institutes agree to the conditions of access. Additional information can be requested from the corresponding author.

## Materials availability

The methods for generating unique biological material used in this study are described in the "Methods" section. Materials are available upon reasonable request from the corresponding author.

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

## Acknowledgements

M.A.B. Ph.D. scholarship funding was courtesy of the USAPA. M.A.B. and D.G.T. were supported by Singapore Ministry of Health's National Medical Research Council (Singapore Translational Research (STaR)) Investigator Award STaR 18nov-0002, the Singapore Ministry of Education under its Research Centres of Excellence initiative, and National Institutes of Health, National Cancer Institute (NIH/NCI) Grant R35CA197697 and NIH/NHLBI P01HL131477-01A1. B.A.B. was supported by the Blavatnik Family Foundation and NIH training grant 5T32CA9302-40. This work was also supported by the NIH/NCI Grant 1R01CA251331 (R.M.), the Ludwig Institute for Cancer Stem Cell Research and Medicine (R.M.) and the Leukaemia and Lymphoma Society Translational Research Programme grant 6619-21 (R.M. and D.T.), co-funded by Snowdome Foundation and the Leukaemia Foundation of Australia. R.M. is a Leukaemia & Lymphoma Society Scholar. D.T. is supported by a Commonwealth Serum Laboratories Centenary Fellowship, Australian Medical Research Future Fund Stem Cells Mission (2008972),

National Health and Medical Research Council Australia Ideas Grants 1182564 & 1184485, Beat Cancer Infrastructure Grant (IF1320) and The Hospital Research Foundation (C-PJ-173-Exper-2019). This project was funded by the Royal Adelaide Hospital Contributing Haematologists' Committee (I.D.L., R.J.D., and S.E.S.), the National Health and Medical Research Council Australia (GNT1047129, R.J.D., T.J.G., A.L.B., I.D.L.) and through the financial and other support of Cancer Council SA's Beat Cancer Project (2002137, R.J.D., D.M.R., and T.J.G.) on behalf of its donors and the State Government of South Australia through the Department of Health. We thank all patients who gave samples for this study, and the South Australian Cancer Research Biobank (SACRB) for their assistance with sample collection and storage. Metabolomics analyses was conducted by Metabolomics Australia at The University of Melbourne, a NCRIS initiative under Bioplatforms Australia Pty Ltd.

## Author contributions

M.A.B. and S.E.S. performed experiments, analysed the data, and wrote the manuscript. K.L. performed ATP rate assays and analysed data. M.A.B. performed the protein structural prediction and modelling. B.A.B. performed analysis of co-occurring complex I variants and IDH mutants. S.B. performed analysis of cell line models. S.K. performed inhibitor experiments with primary cells, 2HG experiments in THP-1 cells and generated co-mutation plots. P.L. analysed NGS data. J.T. analysed gene expression data. C.T.P. and T.N. performed and analysed experiments. K.Z.Y.M., D.A.C., D.G.I. contributed to variant analysis and patient annotation. I.S.P. and D.M.R. designed the mitochondrial copy number assay. J.A.P. and S.M.P. contributed patient material and annotation. S.N. and U.R. performed metabolomic analysis. R.J.D., A.B., D.G.I., I.L., and D.M.R. managed the AML patient database. D.G.T. contributed resources and critical review. N.R. designed experiments and provided critical review. D.M.R. provided clinical information and reviewed the manuscript. R.M. developed and provided critical reagents and resources, and designed experiments for metabolic targeting of IDH1. T.J.G. initiated and oversaw the exome sequencing and edited the manuscript. D.T. designed experiments analysed and interpreted data, conceptualized metabolic implications and reviewed the manuscript. R.J.D. conceptualized the study, analysed and interpreted data, and wrote the manuscript.

## Competing interests

R.M. is on the Board of Directors of CircBio Inc., and Advisory Boards of Kodikaz Therapeutic Solutions Inc. and Syros Pharmaceuticals. R.M. is an equity holder and founder of CircBio Inc. and Pheast Therapeutics Inc.. R.M. is an inventor on several patents related to CD47 cancer immunotherapy licensed to Gilead Sciences, Inc. that are not directly related to the research in this study. The remaining authors declare no competing interests.
