## [Peer Review File · Nature Communications]

Germline mutations in mitochondrial complex I reveal genetic and targetable vulnerability in IDH1-mutant Acute Myeloid LeukaemiaEditorial Note: This manuscript has been previously reviewed at another journal that is not operating a transparent peer review scheme. This document only contains reviewer comments and rebuttal letters for versions considered at *Nature Communications*.

Reviewers' Comments:

Reviewer #1:

Remarks to the Author:

All of my original concerns have been addressed.

I think it would be helpful to comment in the results or discussion that in the models systems tested, the mutant did not change complex I activity. While the technical challenges to the assay are appreciated, the lack of change in complex I activity may indicate that the impairment in respiratory chain function is more subtle.

Reviewer #3:

Remarks to the Author:

the authors answered the questions

Reviewer #4:

Remarks to the Author:

The revised manuscript was substantially expanded to address all reviewers' comments and, as a result, greatly enhanced. Specifically, the authors have adequately addressed all of my previous questions and comments.

The findings of this study demonstrated how germline variants and somatic mutations could function concertedly to shape the genetic underlying of tumor progression and treatment resistance of AML specifically and cancer in general. It is an important study. Nicely done.

Response to Reviewers
Nature Communications manuscript NCOMMS-21-40302-A

We thank the reviewers for their thorough review of our manuscript and acknowledge their contribution to improving the quality and impact of this work.

Reviewer #1 (Remarks to the Author):

All of my original concerns have been addressed.

I think it would be helpful to comment in the results or discussion that in the models systems tested, the mutant did not change complex I activity. While the technical challenges to the assay are appreciated the lack of change in complex I activity may indicate that the impairment in respiratory chain function is more subtle.

Response: We agree that this comment is important in providing context to the experimental result in question. We have included a statement at the end of the results section titled 'Functional analysis of a rare complex I variant in NDUFS8'.

Reviewer #3 (Remarks to the Author):

The authors answered the questions

Reviewer #4 (Remarks to the Author):

The revised manuscript was substantially expanded to address all reviewers' comments and, as a result, greatly enhanced. Specifically, the authors have adequately addressed all of my previous questions and comments. The findings of this study demonstrated how germline variants and somatic mutations could function concertedly to shape the genetic underlying of tumor progression and treatment resistance of AML specifically and cancer in general. It is an important study. Nicely done.